# A First Investigation into the Use of Differential Somatic Cell Count as a Predictor of Udder Health in Sheep

**DOI:** 10.3390/ani13243806

**Published:** 2023-12-10

**Authors:** Marco Tolone, Salvatore Mastrangelo, Maria Luisa Scatassa, Maria Teresa Sardina, Silvia Riggio, Angelo Moscarelli, Anna Maria Sutera, Baldassare Portolano, Riccardo Negrini

**Affiliations:** 1Department of Agricultural Food and Forest Sciences, University of Palermo, Viale delle Scienze, 90128 Palermo, Italy; salvatore.mastrangelo@unipa.it (S.M.); mariateresa.sardina@unipa.it (M.T.S.); silvia.riggio01@unipa.it (S.R.); baldassare.portolano@unipa.it (B.P.); 2Istituto Zooprofilattico Sperimentale della Sicilia, Via Gino Marinuzzi 3, 90129 Palermo, Italy; luisa.scatassa@izssicilia.it; 3Istituto Sperimentale Zootecnico per la Sicilia, Via Roccazzo 85, 90136 Palermo, Italy; angelo.moscarelli@gmail.com; 4Department of Chemical, Biological, Pharmaceutical, and Environmental Sciences, University of Messina, 98166 Messina, Italy; asutera@unime.it; 5Associazione Italiana Allevatori, Via Tomassetti Giuseppe 9, 00161 Rome, Italy; negrini.r@aia.it; 6Department of Animal Science, Food and Nutrition—DIANA, University Cattolica del Sacro Cuore, Via Emilia Parmense 84, 29122 Piacenza, Italy

**Keywords:** differential somatic cell count, mammary gland, receiver-operating characteristic curve, sheep, somatic cell count

## Abstract

**Simple Summary:**

To determine the use of differential somatic cell count as an indicator of mammary health in sheep, the receiver-operating characteristic curve analysis approach was used to find a threshold value that can be used to discriminate healthy animals from those with a probable intramammary infection. This threshold will enhance the precision of diagnosing mammary gland inflammation in dairy sheep.

**Abstract:**

Differential somatic cell count (DSCC), the percentage of somatic cell count (SCC) due to polymorphonuclear leukocytes (PMNs) and lymphocytes (LYMs), is a promising effective diagnostic marker for dairy animals with infected mammary glands. Well-explored in dairy cows, DSCC is also potentially valid in sheep, where clinical and subclinical mastitis outbreaks are among the principal causes of culling. We pioneered the application of DSCC in dairy ewes by applying receiver-operating characteristic (ROC) curve analysis to define the most accurate thresholds to facilitate early discrimination of sheep with potential intramammary infection (IMI) from healthy animals. We tested four predefined SCC cut-offs established in previous research. Specifically, we applied SCC cut-offs of 265 × 10^3^ cells/mL, 500 × 10^3^ cells/mL, 645 × 10^3^ cells/mL, and 1000 × 10^3^ cells/mL. The performance of DSCC as a diagnostic test was assessed by examining sensitivity (Se), specificity (Sp), positive predictive value (PPV), negative predictive value (NPV), and area under curve (AUC) analyses. The designated threshold value for DSCC in the detection of subclinical mastitis is established at 79.8%. This threshold exhibits Se and Sp of 0.84 and 0.81, accompanied by an AUC of 0.88. This study represents the inaugural exploration of the potential use of DSCC in sheep’s milk as an early indicator of udder inflammation.

## 1. Introduction

Small ruminants, such as sheep and goats, contribute not only to the economic well-being of local communities through meat and dairy production but also aid in landscape management, preventing wildfires, and maintaining biodiversity. Italy has about 5.9 million sheep, whose milk is used principally to produce dairy products (Sistema informativo veterinario https://www.vetinfo.it/j6_statistiche/#/report-pbi/89 accessed on 25 November 2023). In Sicily, sheep farming accounts for 12% of the total Italian sheep population, and the most common breed is the Valle del Belice, with around 160,000 head.

Mastitis remains the most widespread disease impacting dairy sheep, posing a threat to animal welfare, and causing significant economic losses [1]. Riggio et al. [2] estimated the base cost of mastitis of € 50.00 split into veterinary, discarded milk and farmer’s time. According to the data from the experimental Zoo Prophylactic Institute of Sardinia, the prevalence of clinical and subclinical mastitis in sheep in Italy varies depending on the region and the farming system. In general, it is estimated that the prevalence of clinical mastitis cases is between 2% and 10% of lactating sheep. The prevalence of subclinical mastitis cases is much higher and can reach 50% or more [3]. Resistance to mastitis is a complex and multifactorial trait depending on genetic (low heritability) and environmental factors, including infection pressure [4]. Assessing the genetic basis of mastitis resistance is challenging due to the trait’s threshold nature: it is discretely expressed in a limited number of phenotypes (typically binary), but it is based on an underlying continuous distribution of factors contributing to the trait referred to as underlying liability [5,6]. Selection for genetic resistance to mastitis can be direct through the diagnosis of the infection (e.g., bacteriological examination of milk and observation of clinical cases of mastitis), or indirect. The microbiological analysis of milk samples to identify the presence of intramammary infections is not applicable as a routine monthly examination in conjunction with the dairy herd improvement milk test due to its cost and demanding lab protocol [7]. A robust indirect proxy for selecting against mastitis is somatic cell count (SCC) in milk, a well-known indicator of udder health and milk quality in dairy animals. The estimated genetic correlations between the udder infection status and SCC indicate that the negative selection for SCC will potentially reduce the incidence of mastitis [8,9].

In cattle, it is generally considered that mammary glands are healthy when the SCC is below 100 × 10^3^ cells/mL, and the threshold to differentiate between healthy and infected cows is set at 200 × 10^3^ cells/mL [10]. However, in the case of sheep, there are currently no widely agreed-upon thresholds established [2,11]. A research effort carried out by a consortium of Italian zoo prophylactic institutes has pinpointed a distinctive threshold in bacteriological isolation, measuring at 265 × 10^3^ cells/mL, effectively distinguishing between positive and negative animals [3]. Other investigations have proposed a crucial upper boundary of 500 × 10^3^ cells/mL for maintaining udder health, as indicated by other authors [12,13]. In a previous study, Riggio et al. [2] applied receiver-operating characteristic (ROC) curve analysis to find an SCC threshold in dairy ewes using bacteriological analysis as a reference standard. The authors recommended a threshold value of 645 × 10^3^ cells/mL for discriminating between the presence or absence of intramammary infection (IMI) in sheep. In the United States, the Food and Drug Administration has established specific legal limits for somatic cell counts (SCCs) in milk. For cows, the limit is set at 750 × 10^3^ cells/mL, while for goats and sheep, it is set at 1000 × 10^3^ cells/mL, as noted by Paape et al. [14]. Schwarz et al. [15] confirmed that a quarter foremilk threshold of 100 × 10^3^ cells/mL effectively distinguishes between infected and non-infected mammary glands. However, they also identified an 8.5% prevalence of mastitis pathogens in mammary glands with somatic cell counts (SCCs) ranging from 1000 to 100,000 cells/mL [14].

It is also important to note that while total SCC in milk is informative for assessing the inflammatory status of the mammary gland, it does not provide a clear breakdown of the proportion of different cell types, such as polymorphonuclear leukocytes (PMNs), macrophages (MACs), lymphocytes (LYMs), and various epithelial cells [16]. These cell types have distinct roles in the immune response to mastitis, and their quantities can vary depending on the stage of infection and the overall health of the mammary gland [17]. The novel generation of the high-throughput flow-cytometry-based analyzer (i.e., Fossomatic 7 DC, FOSS, Hillerød, Denmark) allows counting immune cell populations. Specifically, the differential somatic cell count (DSCC) parameter is calculated as the sum of PMNs and LYMs expressed as a percentage of total SCC. The proportion of macrophages (MACs) can then be determined by subtracting the DSCC value from 100 (MAC = 100 − DSCC).

It is worth noting that DSCC has been suggested as a highly sensitive indirect biomarker for mastitis, capable of detecting inflammatory conditions even when the total somatic cell count (SCC) is low [15]. For example, in the early stages of mastitis, SCC can be relatively low, around 100,000 cells/mL, whereas the proportion of PMNs can be very high, as high as 90% of the SCC. Multiple research investigations in dairy cows have already demonstrated that DSCC serves as a promising phenotypic indicator for selecting against udder health issues [18,19]. Kirkeby et al. [20] discovered that in addition to SCC, DSCC provides a significantly improved indication of intramammary infection (IMI) compared to SCC alone. The study also revealed associations between DSCC and DIM, parity, and pathogen group [20].

As far as we are aware, there has been limited exploration of DSCC in the context of dairy sheep. In an effort to address this gap, we aim to establish a DSCC threshold that can effectively distinguish between sheep at risk of intramammary infection and those that are healthy by examining various cutoff values of SCC.

## 2. Materials and Methods

### 2.1. Milk Sample Collection

A total of 5874 milk samples were collected from 60 Sicilian farms that adhere to the monthly individual DHI performance recording, spanning from March 2021 to January 2022. The sampling procedures strictly adhered to European regulations (Council Regulation (EC) No. 1/2005 and Council Regulation (EC) No. 1099/2009). The study protocol obtained approval from the Bioethics Committee at the University of Palermo (Protocol code UNPA-CLE–98597). In this study, sheep of the Valle del Belice enrolled in the breed herd book were selected from herds with a size representative of Sicilian farming system. Flocks were managed under extensive or semi-extensive grazing systems with supplementary concentrates provided as needed. Within the flock, lactating ewes with no clinical signs of mastitis were eligible for inclusion in the study. No information on previous cases of mastitis in the animals considered is available for this study. Milk samples from both mammary halves (volume: 50 mL) were aseptically collected after routine cleaning and disinfection of the udder and discarding of the first streaks of milk. Every milk sample, without preservatives, underwent rapid cooling and same-day transportation in iceboxes directly to the ISZ milk laboratory for the determination of somatic cell count (SCC) and differential somatic cell count (DSCC), as outlined in the following description.

### 2.2. Analysis of SCC and DSCC

SCC and DSCC were determined with Fossomatic 7 DC (FOSS, Hillerød, Denmark). The working principle of the instrument is based on the staining of cell nuclei and other compartments with acridine orange fluorescent dye, after which they are counted electronically. Briefly, DSCC analysis with a Fossomatic DC instrument involves two steps; in the first one, fluorescence emission from channels 1 and 2 are used for SCC determination; then, cells are further investigated using an algorithm based on a dot plot to distinguish MACs and PMNs together with LYMs [17].

The reliability of SCC and DSCC measurements is assessed through a parameter known as Good Separation (Gose). A Gose value of 1 means that the analysis of a milk sample is considered trustworthy as SCC and DSCC are correctly separated from the background. Conversely, if the Gose value registers at 0, the analyses are not reliable [21].

Prior to conducting the analysis, the instrument underwent a thorough inspection in accordance with the manufacturer’s guidelines. This included the blank sample test to ensure the cleanliness of the flow cell and the utilization of an FMA DC adjustment sample to verify the proper alignment of the laser and detectors. All samples were preheated in a water bath at a temperature of 40 ± 2 °C for a duration of 15 min prior to the analysis.

### 2.3. Statistical Analysis

The dataset included information on herds, ewe ID, SCC, and DSCC. We excluded observations with SCC values below 50,000 cells/mL, as DSCC can exhibit significant variation below this value, as discussed in detail by Damm et al. [17]. Additionally, observations were discarded if the Gose value equaled zero.

To determine the presence or absence of subclinical mastitis we tested four predefined SCC cut-off values established in previous research. Specifically, we applied SCC cut-offs of:(a)265 × 10^3^ cells/mL [3];(b)500 × 10^3^ cells/mL [12,13];(c)645 × 10^3^ cells/mL [2];(d)1000 × 10^3^ cells/mL [14].

To identify the most accurate DSCC cut-off value, we conducted data analysis using the ROC curve and employed the OptimalCutPoints package developed in R [22]. We applied the Youden method, which determines the cut-off point where the sum of sensitivity and specificity is maximized. The prevalence of mastitis was estimated by considering the number of milk samples that exceeded the SCC cut-off values divided by the total number of observations.

## 3. Results

This research encompassed the analysis of DSCC and SCC data derived from 5874 individual milk samples. The summary statistics in Table 1 display the mean values along with their standard errors (SE) for DSCC across various SCC categories. These values exhibited a range from 61.13% to 85.75% suggesting a higher percentage of PMNs + LYMs compared to MACs in the mammary glands of the ewes.

Notably, as SCC by class levels increased (with SCC classes ranging from 50,000 cells/mL to 1,500,000 cells/mL), DSCC levels also showed an upward trend. The proportion of DSCC among the SCC classes differed significantly (*p* < 0.01).

When considering different SCC thresholds, the prevalence of mastitis varied. As expected, it ranged from a minimum of 26.5% for the cut-off of 1000 × 10^3^ cells/mL to a maximum of 52.8% for a cut-off of 265 × 10^3^ cells/mL.

The results of the ROC analyses, aimed at determining the most suitable cut-off point for DSCC, while considering different SCC thresholds for distinguishing between healthy animals and those with mastitis are reported in Table 2.

The AUC-ROC (Area Under Curve-ROC) metric assesses the performance of a classification model across various threshold configurations, examining both probability curves and separability measures. The estimated ROC curve and statistics for the diagnostic test using SCC > 265 × 10^3^ cells/mL are shown in Figure 1A and Table 2. The optimal threshold for DSCC was determined to be 76.1%, yielding an AUC of 0.81 (*p* < 0.05). The sensitivity and specificity were 0.72 and 0.77, respectively. The positive predictive value (PPV) and negative predictive value (NPV) were 0.77 and 0.72, respectively. Figure 1B and Table 2 show the ROC curve and statistics considering a cut-off of 500 × 10^3^ of SCC. The best threshold for DSCC was 78.7%, with an AUC of 0.86, and both Se and Sp were 0.80. The PPV and NPV were 0.71 and 0.87, respectively. Considering a cut-off value of 645 × 10^3^ of SCC, a threshold for DSCC of 79.8% was obtained (Figure 1C and Table 2). At this value, the test had an AUC of 0.89, with Se and Sp values of 0.84 and 0.81, respectively. PPV and NPV were 0.69 and 0.91, respectively. Finally, the estimated ROC curve and statistics for the diagnostic test using SCC > 1000 × 10^3^ are shown in Figure 1D and Table 2. The optimal threshold for DSCC was 79.8%, with an AUC of 0.89. At this threshold for DSCC, Se and Sp values were 0.90 and 0.77, with a PPV and NPV of 0.58 and 0.95, respectively.

## 4. Discussion

In recent times, there has been a growing demand within the dairy industry to decrease the use of antibiotics at the herd level and to reserve antimicrobial treatments exclusively for animals confirmed to be truly affected by mastitis. This shift is driven by the imperative to face antibiotic resistance in animals and to ensure the safety of consumers of dairy products. Currently, the sole diagnostic test with the capacity to robustly distinguish between infected and healthy animals is the bacteriological examination. Nevertheless, its practical implementation as a routine test is hindered by its cost and time-intensive protocol. Biomarkers—like SCC and DSCC—are often used in health screening. For this purpose, it is necessary to identify the optimal cut-off to correct cluster positives (sick animals) or negatives (healthy animals) [23]. With this aim and for the first time in dairy sheep, we conducted a ROC analysis to estimate a DSCC threshold. This threshold enhances the precision of diagnosing mammary gland inflammation in dairy sheep.

In this retrospective analysis, the rate of subclinical mastitis occurrence was determined by the ratio between the animals that surpassed the predefined SCC threshold and the total number of observations. Assuming an SCC cut-off of 265 × 10^3^, the rate of subclinical mastitis incidence stood at 52.8%. This figure is notably elevated when compared to the rates typically documented in the literature for dairy sheep farms [24]. According to Contreras et al. (2007), the incidence of subclinical mastitis in small ruminants averages 5–30% [25]. With a SCC threshold of 500 × 10^3^ cells/mL, to distinguish between healthy and infected animals, we observed a prevalence of 38.7%. This figure aligns closely with the findings of Tolone et al. [5], who, in the same sheep breed, reported a prevalence of 38% using a reference standard to identify infected animals. However, it exceeds the figures reported by Pengov et al. [11] and Gonzalo et al. [26], which were 26.2% and 24.5%, respectively. Such incidence values are similar to those identified in the present study using a higher cut-off (1000 × 10^3^ cells/mL). Ruegg et al. [27] reported an incidence rate for subclinical mastitis in dairy sheep ranging from 15 to 30%.

The test characteristics of DSCC were evaluated by comparing Se, Sp, PPV, and NPV, as well as AUC analyses quantifying the overall ability of a test to discriminate between healthy and diseased individuals.

An ideal diagnostic correctly identifies all animals that truly have the disease (Se = 100%), while also accurately distinguishing those that are healthy (Sp = 100%). However, the reality is that such a perfect test does not exist. Consequently, it becomes crucial to establish an optimal threshold that strikes a balance between Se and Sp, maximizing their combined effectiveness.

It is important to clarify that, here, we assume equal importance between sensitivity and specificity, but this assumption may not always hold since the significance of these two parameters depends on the test’s specific objectives.

In an effort to minimize the erroneous administration of antibiotics to uninfected ewes, one could opt for a threshold with elevated Sp and PPV. However, this choice comes at the expense of potentially treating infected ewes inadequately. To strike a balance between the reduction in antibiotic use and the economic losses caused by mastitis, it is essential to maximize the Sp and PPV of the tests while simultaneously minimizing any decrease in Se and NPV.

Considering these factors, the designated threshold value for DSCC in the detection of subclinical mastitis is established at 79.8%. This threshold exhibits a Se and Sp of 0.84 and 0.81, respectively, accompanied by an impressive AUC of 0.88. The PPV and NPV play a significant role in assessing a diagnostic test’s capacity to accurately determine the presence or absence of the condition in question. Our method’s PPV stands at 0.69, indicating a 69% probability of correctly identifying individuals with the condition of interest when the test outcome is positive. Despite this, the test offers value due to its affordability and convenience. The true strength of our screening test, however, lies in its NPV, providing us with a 91% level of confidence that a subject testing negative does not have the condition of interest. This discourages the unnecessary use of preventive antimicrobial treatments.

Comparable data on DSCC in dairy ewes are currently not present in the scientific literature, given the novelty of the parameter. However, while our results pertain to a different species, they align with findings from Damm et al. [17], who reported DSCC percentages ranging from 72.68% to 76.12% in cattle. In a similar study, Bobbo et al. [18] demonstrated mean values around 65% for cattle. These percentages in cattle typically indicate the presence of subclinical infections. Notably, Dal Prà et al. [28] recently observed an increase in DSCC from 56% to 65%, with values peaking at 70–72%, as SCC ranged from >200 × 10^3^ cells/mL to over 1000 × 10^3^ cells/mL. In a prior study involving Sarda sheep, the authors reported mean PMN values around 30–40% when SCC concentrations were below 100 × 10^3^ cells/mL [29]. Conversely, with SCC levels exceeding 400 × 10^3^ cells/mL, PMNs averaged around 70%. While physiological and environmental factors exert a minor influence on PMN quantities, this notable increase in PMNs is likely associated with an enhanced immune defense, particularly when the mammary gland necessitates recovery from existing infections and protection against the onset of new ones [30]. In Holstein pluriparous cows, DSCC thresholds demonstrating optimal performance ranged from 71.2% to 74.5%, with AUC values surpassing 0.85 [31]. Finally, Schwarz et al. [19], using a cutoff of 200 × 10^3^ cells/mL for SCC alone or in combination with DSCC (thresholds > 200 × 10^3^ cells/mL and >60%, respectively), observed a variation of 19% in Se (from 47% to 66%) and a decrease of 20% change in Sp (from 74% to 54%).

## 5. Conclusions

To the best of our knowledge, this study represents the inaugural exploration of the potential use of DSCC in sheep’s milk as an early indicator of udder inflammation. While species-specific comparisons are lacking in the scientific literature, our findings, when compared with those from studies involving cows, exhibit promise.

In our next step, we will include bacteriological analysis as the gold standard for categorizing animals as infected or not infected following the entire lactation of the animals under study.

## Figures and Tables

**Figure 1 animals-13-03806-f001:**
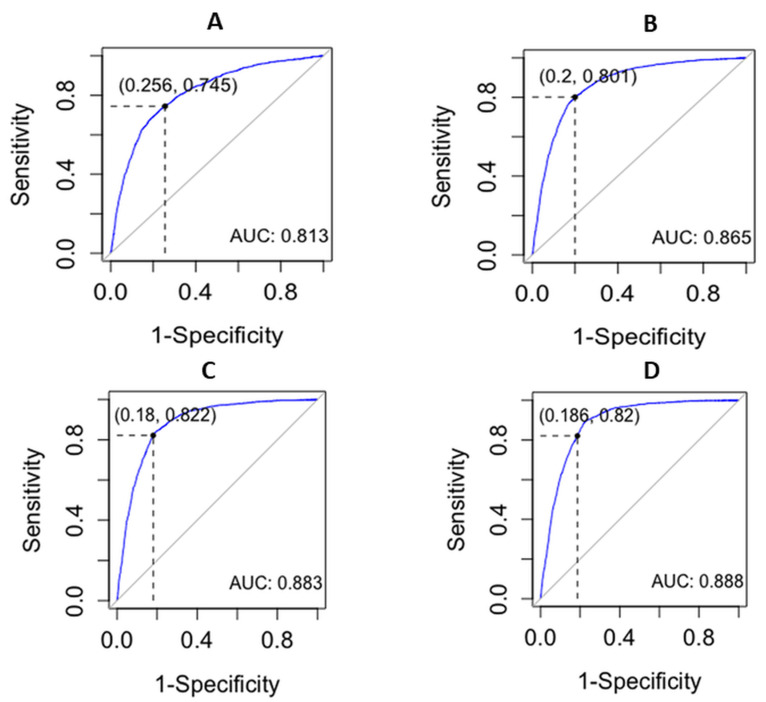
Receiver-operating characteristic curve (ROC) illustrating the performance of differential somatic cell count (DSCC) in identifying intramammary infection at (**A**) 265 × 10^3^ cells/mL; (**B**) 500 × 10^3^ cells/mL; (**C**) 750 × 10^3^ cells/mL; and (**D**) 1000 × 10^3^ cells/mL; AUC: area under curve.

**Table 1 animals-13-03806-t001:** Minimum, maximum, mean and standard error (SE) statistics for differential somatic cell count (DSCC) and SE by somatic cell count (SCC) classes.

SCC Class (×1000)	*n*	Min	1st Qu. ^1^	3rd Qu. ^2^	Max	DSCC (%) ^3^	SE
50 < SCC ≤ 250	2690	7.3	47.8	75.3	97.6	61.13 ^a^	0.34
250 < SCC ≤ 500	908	11.7	59.9	79.5	95.3	68.27 ^b^	0.51
500 < SCC ≤ 750	448	27.7	70.8	83.6	95.3	75.20 ^c^	0.59
750 < SCC ≤ 1000	272	19.70	77.2	85.6	93.0	80.14 ^d^	0.58
1000 < SCC ≤ 1500	400	46.60	80.2	87.4	95.60	83.01 ^e^	0.36
SCC > 1500	1156	6.30	83.5	88.8	97.80	85.75 ^f^	0.17

^1^ 1st Qu.: first quartile, ^2^ 3rd Qu.: third quartile, ^3^ Different superscript letters are significantly different at *p* < 0.01.

**Table 2 animals-13-03806-t002:** Results of ROC analysis to identify the best cut-off for DSCC according to different SCC thresholds identified in previous studies.

SCC (Cells/mL)	0 vs. 1 ^1^	Threshold %	Se	Sp	PPV	NPV	AUC
265K	2770–3104	76.1	0.72 (0.71–0.74)	0.77 (0.75–0.79)	0.77 (0.76–0.79)	0.72 (0.70–0.74)	0.81 (0.80–0.82)
500K	3598–2276	78.7	0.80 (0.78–0.82)	0.80 (0.79–0.81)	0.71 (0.69–0.73)	0.87 (0.85–0.88)	0.86 (0.85–0.87)
645K	3898–1976	79.8	0.84 (0.82–0.85)	0.81 (0.80–0.82)	0.69 (0.67–0.71)	0.91 (0.90–0.91)	0.88 (0.87–0.89)
1M	4318–1556	79.8	0.90 (0.88–0.91)	0.77 (0.76–0.78)	0.58 (0.57–0.62)	0.95 (0.94–0.96)	0.89 (0.88–0.90)

^1^ 0: healthy, 1: infected; Se: sensitivity, Sp: specificity, PPV: positive predictive value, NPV: negative predictive value, AUC: area under curve; 95% Confident interval between the brackets.

## Data Availability

Research data will be provided upon request.

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
