# Peer review of "A First Investigation into the Use of Differential Somatic Cell Count as a Predictor of Udder Health in Sheep"

_animals, 2023, doi:10.3390/ani13243806_

Round 1
Reviewer 1 Report
Comments and Suggestions for Authors
The aim of the study has serious practical impact. The mastits in sheep is not as widely studied as problem of dairy cows' mastitis.
The Reviewer would suggest changing the title of the manuscript - eg. Innovative approach to evaluation of ....
"Mammary gland" should be added to the keywords list.
lines 71-72 why Authors didnt mentioned limits from UE Reg. 853/2004?
The introduction part presents good background for the study, and analysis of different approaches to SCC. The reviewer would suggest suplementing the information about the situation of sheep in Italy, statistics of mammary gland inflammation problem and economical loses.
The number of farms enrolled into the study and number of collected milk samples is sufficient.
M&M part: lacks in data from the farms - breed of sheep, number of lactation of lactating ewes, milk yield, number of parturitions, previous incidents of mastitis on farms, vet health check.
Would be good to add direct method of methylene blue staining& modified California mastitis test or Whiteside's test on farm to estimate leucocytes. And Giemsa and May-Grunwald staining to distinguish lecocyte types.
There is no need to double presentation of the results on Figs and in table.
The results are presented very briefly.
line 236 it is not only the aspect of animal welfare but also economy - cost of vet treatment& milk loses.
With the lack of variables from farm data its hard to evaluate discussion findings.
Comments on the Quality of English Languagethe reviewer does not see serious language faults
Author Response
Reviewer 1
The aim of the study has serious practical impact. The mastits in sheep is not as widely studied as problem of dairy cows' mastitis.
The Reviewer would suggest changing the title of the manuscript - eg. Innovative approach to evaluation of ....
Thank you for your suggestion. The authors are inclined to keep this title as is a preliminary study on the use of DSCC to detect mastitis in sheep. This study will be validated by considering the bacteriological examination as the reference method for the diagnosis of mastitis and the data related to the animal.
"Mammary gland" should be added to the keywords list.
Done
lines 71-72 why Authors didnt mentioned limits from UE Reg. 853/2004?
Thanks for the comment .The UE Reg. 853/2004 does not report SCC specific limit for sheep milk but only the CFU limit for species other than cows.
The introduction part presents good background for the study, and analysis of different approaches to SCC. The reviewer would suggest suplementing the information about the situation of sheep in Italy, statistics of mammary gland inflammation problem and economical loses.
Thanks for the suggestion. The required information has been added in the introduction (lines 43-56) of the revised manuscript.
The number of farms enrolled into the study and number of collected milk samples is sufficient.
M&M part: lacks in data from the farms - breed of sheep, number of lactation of lactating ewes, milk yield, number of parturitions, previous incidents of mastitis on farms, vet health check.
Thanks for the comment. We added the available information in M&M section. Unfortunately, as described below, this is a retrospective study, and some information are not available.
Would be good to add direct method of methylene blue staining& modified California mastitis test or Whiteside's test on farm to estimate leucocytes. And Giemsa and May-Grunwald staining to distinguish lecocyte types.
In this retrospective study, we did not use CMT as the gold standard. Additionally, milk samples were collected from animals that did not exhibit obvious signs of mastitis. As reported by several authors, the usefulness of CMT as indicator of subclinical mastitis is still unclear; healthy sheep normally have a higher SCC than cows, making the interpretation of the CMT harder (Maisi et al., 1987; Fthenakis et al., 1991; González-Rodríguez et al., 1995). Moreover, as reported by Riggio et al. (2013) the CMT appeared only to discriminate udders infected with major pathogens.
There is no need to double presentation of the results on Figs and in table.
In possible, to improve the clarity of the results presentation, the authors prefer to keep both the table and the graphs.
The results are presented very briefly.
line 236 it is not only the aspect of animal welfare but also economy - cost of vet treatment& milk loses.
Thank you for your suggestion. We added this information at line 254 of the revised manuscript.
With the lack of variables from farm data its hard to evaluate discussion findings.
Thank you for your comments. The information regarding the animals' lambing dates, lactation duration, or other animal-specific details are not available in this retrospective study. This study represents the inaugural exploration of the potential use of DSCC in sheep’s milk as an early indicator of udder inflammation. In our next step, we will include bacteriological analysis as the gold standard for categorizing animals as infected or not infected following the entire lactation of the animals under study.
We hope that our revisions, responses and considerations will now meet with your approval.
Best regards,
Reviewer 2 Report
Comments and Suggestions for Authors
Dear authors,
I hope this letter finds you well. I have carefully reviewed your article titled "A first investigation into the use of differential somatic cell 2 count as a predictor of udder health in sheep," and I appreciate the effort you have put into your research. However, I would like to bring some important points to your attention for further consideration and clarification.
Diagnostic Methodology for Mastitis:
It is unclear from the article how the final diagnosis of mastitis was determined. It would be beneficial for readers if you provide a detailed explanation of the diagnostic methods employed and the criteria used to establish the diagnosis. Clarity on this aspect will enhance the robustness of your study.
Omission of Bacteriological Examinations:
I noticed that there is no mention of bacteriological examinations in your study. Given the significance of identifying the causative agents of mastitis, it would be valuable to explain why bacteriological examinations were not conducted. Providing this information will strengthen the rationale behind your methodological choices.
I understand that some additional comments are provided in the text; however, addressing the above points will further strengthen the clarity and validity of your research. I encourage you to consider these suggestions in your revisions.
Thank you for your time and dedication to advancing scientific knowledge. I look forward to seeing the improvements in your article.
Best regards,

Author Response
Dear authors,
I hope this letter finds you well. I have carefully reviewed your article titled "A first investigation into the use of differential somatic cell 2 count as a predictor of udder health in sheep," and I appreciate the effort you have put into your research. However, I would like to bring some important points to your attention for further consideration and clarification.
Thank you very much for your comments and suggestion.
Diagnostic Methodology for Mastitis:
It is unclear from the article how the final diagnosis of mastitis was determined. It would be beneficial for readers if you provide a detailed explanation of the diagnostic methods employed and the criteria used to establish the diagnosis. Clarity on this aspect will enhance the robustness of your study.
Thanks for the comment. The diagnosis of mastitis is based on threshold values for SCC, which have been identified in previous studies, to discriminate between healthy animals and animals with intramammary infections. This information is reported in the materials and methods section at lines 157-163. In fact, in this retrospective study, milk samples were collected from animals that did not exhibit obvious signs of mastitis.
Omission of Bacteriological Examinations:
I noticed that there is no mention of bacteriological examinations in your study. Given the significance of identifying the causative agents of mastitis, it would be valuable to explain why bacteriological examinations were not conducted. Providing this information will strengthen the rationale behind your methodological choices.
We took in close examination the reviewer’ comment. This is a retrospective study on the use of DSCC to detect mastitis in sheep considering only the SCC. We know that bacteriological analysis is the gold standard to diagnose the mastitis. This manuscript represents the inaugural exploration of the potential use of DSCC in sheep’s milk as an early indicator of udder inflammation. In our next step, we will include bacteriological analysis as the gold standard for categorizing animals as infected or not infected following the entire lactation of the animals under study. In fact, this study will be validated by considering the bacteriological examination as the reference method for the diagnosis of mastitis and the data related to the animal. These considerations were reported in the conclusion section of the manuscript.
I understand that some additional comments are provided in the text; however, addressing the above points will further strengthen the clarity and validity of your research. I encourage you to consider these suggestions in your revisions.
Thanks for the additional comments.
Thank you for your time and dedication to advancing scientific knowledge. I look forward to seeing the improvements in your article.
We hope that our revisions, responses and considerations will now meet with your approval.
Best regards,
Round 2
Reviewer 2 Report
Comments and Suggestions for Authors
Dear Author's
I hope this letter finds you well. I am writing to inform you that your submitted article, titled "A first investigation into the use of differential somatic cell count as a predictor of udder health in sheep " has undergone a thorough review process, and I am pleased to inform you that all recommended changes have been implemented successfully.
The revisions you made have significantly strengthened the clarity, coherence, and overall quality of the article.
Best regards,